# The effect of Mn$_2$Sb$_2$ and Mn$_2$Sb secondary phases on magnetism in (GaMn)Sb thin films

**Jorge A. Calderón**[1☯], **F. Mesa**[2☯]*, **A. Dussan**[1☯], **R. González-Hernandez**[3☯], **Juan Gabriel Ramirez**[4☯]

**1** Departamento de Física, Grupo de Materiales Nanoestructurados y sus Aplicaciones, Universidad Nacional de Colombia-Colombia, Bogotá, Colombia, **2** Universidad del Rosario, Faculty of Natural Sciences, NanoTech Group, Bogotá, Colombia, **3** Departamento de Física, Grupo de Investigación en Física Aplicada, Universidad del Norte, Barranquilla, Colombia, **4** Department of Physics, Universidad de los Andes, Bogotá, Colombia

☯ These authors contributed equally to this work.
* fredy.mesa@urosario.edu.co

**Data Availability Statement:** All relevant data are within the manuscript.

**Funding:** The Universidad Nacional de Colombia - COLCIENCIAS Quipu Code: 201010020958,

## Abstract

In this work, a detailed study of structural, electrical and magnetic characterization of (GaMn)Sb diluted magnetic semiconductors (DMS) is presented. (GaMn)Sb thin films were grown by DC magnetron co-sputtering method as an innovative procedure to fabricate III-V DMS. The presence of unusual Mn$_2$Sb$_2$ and Mn$_2$Sb secondary phases, induced by substrate temperature and deposition time, were revealed through XRD measurements. Magnetization measurements allow determining crossover between a paramagnetic-like to a ferromagnetic-like behavior controlled by secondary phases. It was found that both, the magnetic remanence and magnetic coercivity, increases with substrate temperature. Interestingly, the magnetic response is paramagnetic at lower deposition times and substrate temperatures, and XRD measurements suggest the absence of Mn$_2$Sb and Mn$_2$Sb$_2$ in secondary phases. For longer deposition times or higher substrate temperature, XRD shows the presence of Mn$_2$Sb$_2$ and Mn$_2$Sb phases and ferromagnetic-like behavior. The DC resistivity of our samples was characterized and the carrier density was determined by Hall measurements and, in contrast with the reported in other studies, found them to be a p-type semiconductor with carrier densities as big as one order of magnitude larger than reported values. From the ferromagnetic-like samples, evidence of an anomalous Hall-effect in the sample was found, with higher magnetic saturation and a anomalous Hall conductivity of 2380 S/cm. All the results point to a contribution of the secondary phases to the overall magnetic response of the samples used, and suggest the importance of studying the formation of secondary phases in the growth of DMS, especially, for the case of (GaMn)Sb where Mn ion can have multiple oxidation states.

## Introduction

Recent research has been focused on the fabrication and study of the physical properties of DMS, in order to develop promising materials to obtain spintronic devices [1–7].

Universidad del Rosario and Universidad del Norte supported this work. Jorge Arturo Calderón Cómbita is scholarship PhD of Doctorados COLCIENCIAS Conv. 785 – 2017. J.G.R acknowledge support from FAPA program of Facultad de Ciencias and Vicerrectoria de Investigaciones of Universidad de los Andes, Bogotá, Colombia.

**Competing interests:** The authors have declared that no competing interests exist

Fundamental physical studies, as well as new building blocks for spintronics, could emerge from the ability to synthesize DMS materials[8–11]. However, experimental requirements as long processing times and high costs are required in common DMS synthesis [12], which represents a difficulty to implement it in short-term industrial applications [13]. On the other hand, [14, 15] problems such as low adherence between the compound and the substrate used, non-uniform surfaces, and long deposition times, are present in the current low-cost methods that reduce the quality and durability of the devices. Consequently, an optimal technique to fabricate DMS materials is still lacking.

There is a particular interest in DMS compounds based on III-V semiconductors due to its easy fabrication process and silicon-based architectural similarity, that would be allowed for industrial applications [16–18]. For these reasons, DMS compounds based on III-V semiconductor matrices, doped with Mn ions, have been recently studied [16, 19–21]. A relatively low Curie temperature [22], for Mn ion concentration below 10%, has been shown by Ga$_{1-x}$Mn$_x$As and Ga$_{1-x}$Mn$_x$Sb when grown by Molecular Beam Epitaxy (MBE). The formation of binary phases, by this and other methods, has been reported. It is known that dense concentration of Mn, without the generation of binary phases, rises a DMS with semiconductor properties of higher Curie temperature and quality, meaning a challenge for researchers [23]. However, the rise of electrical and magnetic properties, given by the presence of the secondary phases in these compounds, can be used in optoelectronic applications [24–27].

Additionally, the cluster-stoichiometry effect of the binary phase MnSb on the magnetic properties in (GaMn)Sb thin films was studied by Talantsev et al. [28]. Currently, the ferromagnetic origin in DMS materials, especially in III-V semiconductors, is still on debate. Considering the role of the band structure, the ferromagnetic response was attributed to the exchange coupling among Mn impurities.[22, 29, 30].

Similarly, the Anomalous Hall-effect (AHE) in DMS materials has been reported. AHE sheds light in the origin of the ferromagnetic-like coupling since its accounts for the conductivity signal due to the impurities concentration. In GaMnSb thin films, AHE was reported, and a sign of change in the Hall resistance, attributed to the slope change of the DOS at Fermi level, was found [31]. A discussion about the contribution of inherent secondary phases to the AHE is still lacking, especially since some of these secondary phases show ferromagnetic or ferrimagnetic behavior. Thus, in alternative methods to conventional MBE growth, chemical reactions between the transition element ion and the semiconducting matrix have been used [32–34].

In the present work, the study of crystallographic, morphological, magnetic and electrical properties of (GaMn)Sb thin films, fabricated via DC-magnetron co-sputtering method, is reported. Additionally, secondary Mn$_2$Sb and Mn$_2$Sb$_2$ phases were found and their magnetic properties were determined. A ferromagnetic-like behavior that can be tuned with deposition time or temperature was found. These ferromagnetic contributions were corroborated with AHE measurements. Exclusively in a subgroup of samples with high deposition temperatures, AHE was found. Thus, AHE measurements allow differentiating the type of magnetic contributions in (GaMn)Sb thin films. The proposed sputtering method offers a reliable way to growth DMS materials with a wide range of industrial applications.

## Materials and methods

Through DC magnetron co-sputtering method samples were fabricated; an Ar (99.999%) atmosphere of 2.5x10$^{-2}$ Torr was used and controlled through a flow of 20 sccm. The (GaMn) Sb samples were deposited on soda-lime-type glass substrates, 7 cm away from the targets, using GaSb ((Ga), 36.5 (Sb) 63.5 wt%), and Mn (99.9 wt%). Targets were acquired by

Plasmaterials Ltda. Synthesis parameters, such as time of deposition—$t_d$ (10 and 15 min) and substrate temperature—$T_s$ (373, 423 and 523 K), were systematically varied and established after an optimization process of the fabrication method. Additionally, other synthesis parameters, like GaSb magnetron power—$Pw_{GaSb}$ = 100 W and Mn magnetron power—$Pw_{Mn}$ = 65 W, remained constant. All the samples were subjected to *in situ* annealing processes at 673 K for 2 h. The deposition time was varied to obtain thin film thicknesses, between 160 and 245 nm.

The XRD measurements were carried out with X'Pert Pro diffractometer (PANalytical), equipped with a Cu-Kα source: 1.540598 Å and an X′Celerator detector. The Energy-dispersive X-ray spectroscopy (EDXS) measurements were conducted using a JEOL microscope (JSM 6490-LV model), and AFM Asylum Research MFP 3D Bio microscope was used for morphological characterization, employing the tapping mode. Raman spectrum were performed at room temperature using a laser of λ = 532 nm (Power = 45W), with an exposure time of 1000 ms maintaining a beam configuration perpendicular to the sample, through the 50X optical lens given by Renishaw InVia Raman Microscope. The information processing was performed using 4.4 software Wire. The magnetic properties of the samples were measured in a vibrating sample magnetometer (VSM) on a VersaLab system from Quantum Design. Hall-effect measurements were obtained by using Quantum Design PPMS AC Transport Option. The contacts were made with silver paint using a Van Der Pauw method.

## Results and discussion

Fig 1 shows the XRD patterns for the samples fabricated with synthesis parameters reported in Table 1. The identification of crystalline phases related to deposition time and substrate temperature variation can be observed. The XRD pattern for a sample with $t_d$ = 10 min (black line) evidences the material does not have defined crystalline phases. However, the crystalline

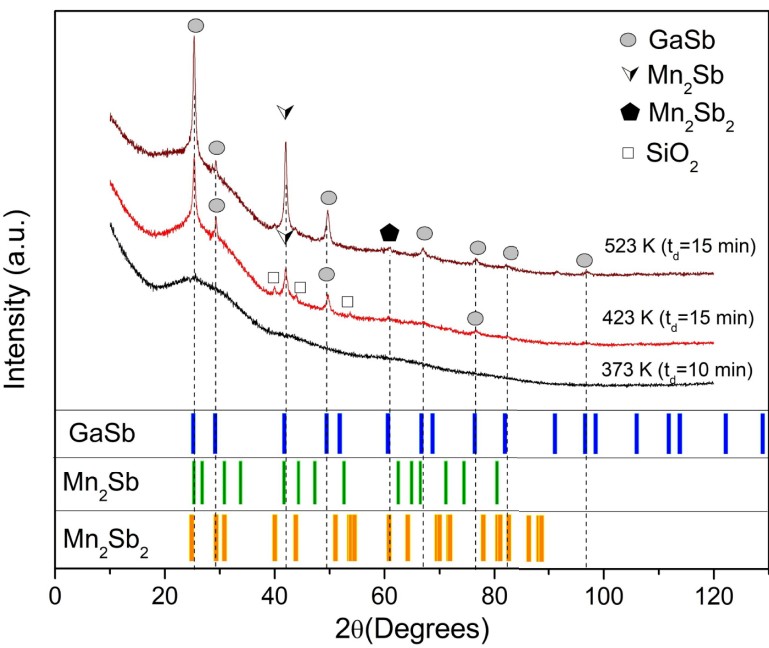

**Fig 1. XRD patterns of GaMnSb compound thin films for $t_d$ = 10 min, $T_s$ = 373 K, $T_s$ = 423 K, and $T_s$ = 523 K with $t_d$ = 15 min.** At the bottom of the figure, a comparison and fitting between the experimental and theoretical base data patterns, solved by the Rietveld method.

**Table 1. Synthesis parameters of GaMnSb thin films.** Report of values of remnant magnetization (M$_r$), coercive field (H$_c$), charge carrier density (n$_p$), Anomalous Hall Coefficient (R$_s$), crystallite size (τ), and EDXS concentration values of elementary composition for a thin film with T$_s$ = 523 K and t$_d$ = 15 min. The values of the experimental error in parentheses were reported.

| Synthesis parameters | T [K] | $M_r$ $10^{-2}$ [emu cm$^{-3}$] | $H_c$ $10^3$ [Oe] | $n_p$ $10^{19}$ [cm$^{-3}$] | | $1/R_sM$ [S/cm] | $\tau$(Å) GaSb / Mn$_2$Sb / Mn$_2$Sb$_2$ |
|---|---|---|---|---|---|---|---|
| t$_d$ = 10 min | 50 | - | - | - | | - | - |
| T$_s$ = 423 K | 150 | - | - | - | | - | - |
| | 300 | - | - | 0.000319 | | - | - |
| t$_d$ = 15 min | 50 | 0.62 (2) | -0.117 (2) | - | | - | 154 / 111 / - |
| T$_s$ = 423 K | 150 | 0.58 (2) | -0.243 (2) | - | | - | |
| | 300 | 0.47 (4) | -0.194 (3) | 2.08 | | - | |
| t$_d$ = 15 min | 50 | 1.30 (2) | -0.214 (2) | I | II | - | 164 / 186 / 274 |
| T$_s$ = 523 K | 150 | 1.29 (3) | -0.364 (1) | | | - | |
| | 300 | 1.18 (2) | -0.315 (6) | 12.1 | 28.6 | 2380 | |

EDXS concentration values of elementary composition for thin film (GaMn)Sb (T$_s$ = 523 K; t$_d$ = 15 min)

| E [kV] | Ga (Wt%) | Mn (Wt %) | Sb (Wt %) | Mn/Ga (%) | Sb/Ga (%) |
|---|---|---|---|---|---|
| 8 | 26.22 | 9.76 | 59.52 | 37.2 | 227.0 |
| 12 | 26.90 | 6.19 | 56.21 | 23.0 | 209.0 |
| 16 | 23.65 | 6.01 | 43.52 | 25.4 | 184.0 |

properties of thin films with t$_d$ = 15 min and different T$_s$ values were studied by using Rietveld refinement (red and wine color lines). Then, the phases identified were GaSb (PDF 00-007-0215), Mn$_2$Sb (PDF 00-004-0822), and Mn$_2$Sb$_2$ (PDF 96-900-8901). These crystal structures correspond to Zinc-Blende, tetragonal and hexagonal, respectively. For all refinements, Rp and Rwp fitting parameters were obtained close to 1,23% and 1,58%, respectively. SiO$_2$ phase is a weak contribute to XRD pattern owing to devitrification tendency, due to the presence of SiO$_2$ nucleation and in consequence of annealing process, as is suggested by others authors [35, 36]; this phase only appears in a sample with T$_s$ = 423 K and t$_d$ = 15 min.

Additionally, EDXS measurements revealed that the relative concentration between Mn and Ga was Mn / Ga = 0.25, for samples at T$_s$ = 423 K with deposition times of 10 and 15 min; whilst for samples deposited at T$_s$ = 523 K (t$_d$ = 15 min) was Mn / Ga = 0.36. EDXS analysis, taken at different incident beam energy, shows that the Mn and Sb elements remain close to the surface of the samples, compared to the presence of Ga over it. Table 1 reports these values.

However, Raman spectroscopy was realized in order to obtain complementary structural information to XRD measurements shown in Fig 1. Fig 2 shows the Raman spectra recorded for GaMnSb samples both, at 423 K and 523 K substrate temperature. The presence of a vibration mode for high frequency region is observed at 647 cm$^{-1}$ when T$_s$ increases (T$_s$ = 523 K); this can be associated to manganese oxides formation on the surface of the sample due to oxidation process after the synthesis of the material. The phonon mode observed in the Raman shift around 650 cm$^{-1}$ has been reported for Mn$_3$O$_4$-like phase [37].

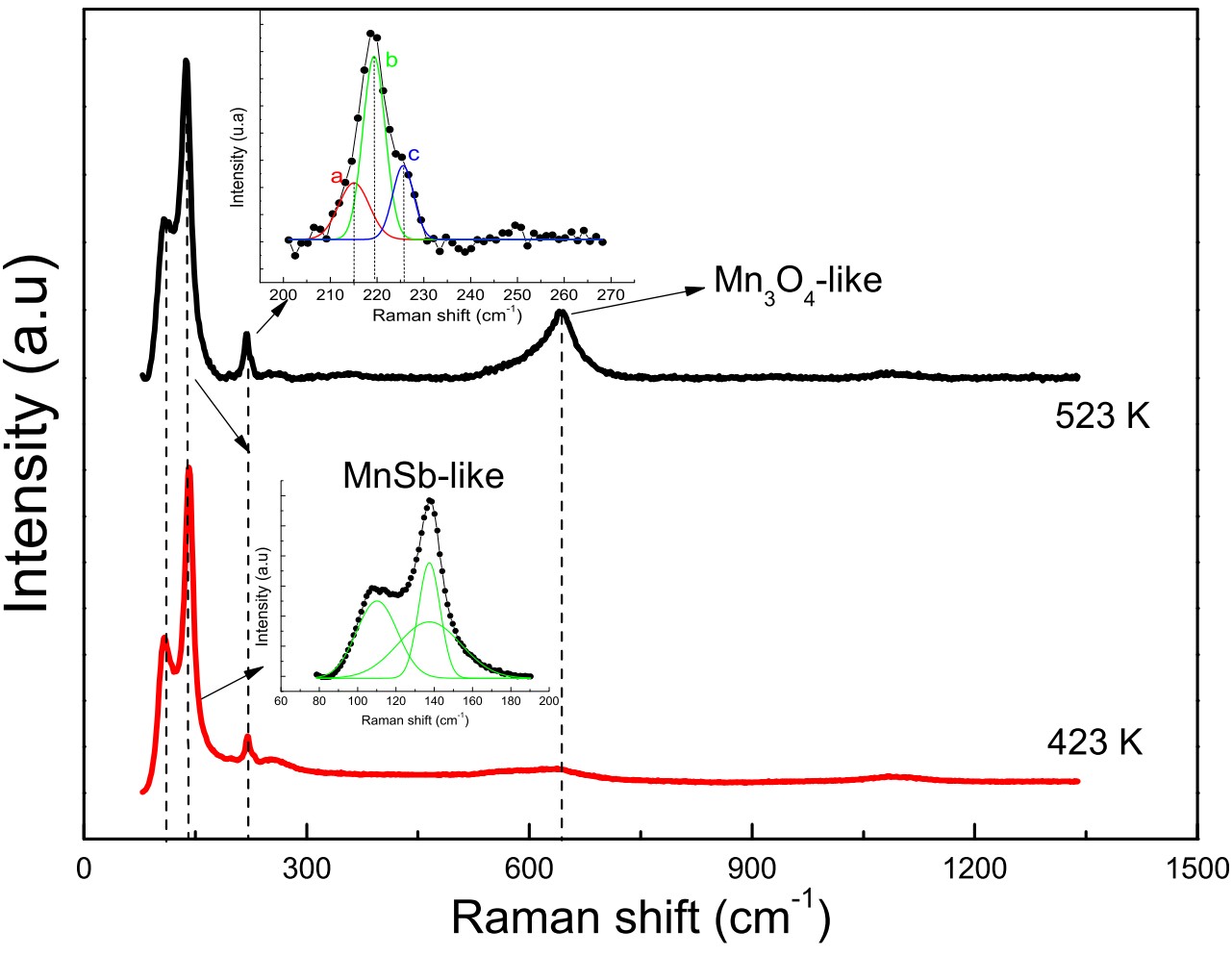

**Fig 2.** Raman shift curves of GaMnSb thin films varying $t_d$ a) 423 K and b) 523 K.

On the other hand, Raman spectra for GaMnSb samples measured show weak phonon modes at 230 cm⁻¹, 141 cm⁻¹ and 108 cm⁻¹, related with LO and TO modes for GaSb and Mn₂Sbx phases on the material. In the up inset of Fig 2 shows the deconvolution of the pick found at 230 cm⁻¹ Raman shift, characterized by three weak vibrations at 226 cm⁻¹, 220 cm⁻¹ and 215 cm⁻¹, which are associated to LO (226 cm⁻¹) and TO modes, respectively. TO modes observed in the samples here, can be related with substrate inhomogeneity and imperfections on surface [38]. The presence of phonon modes observed at 141 cm⁻¹ and 108 cm⁻¹ can be related with the incorporation Mn atoms during synthesis process. It was reported [39] that MnSb-like vibration modes are expected in GaMnSb samples as Mn atoms have taken into GaSb matrix. The above is in accordance with the results reported in Fig 1.

Moreover, magnetic properties of thin films have been studied by measurements of the magnetization response, when the applied magnetic field and temperature were varied. Fig 3A–3C shows the effect of the synthesis parameters on the magnetization (M) behavior as a function of an applied magnetic field (H), whereas the diamagnetic contribution from the glass substrate was discounted. In Fig 3A it is observed a linear correlation at 50 K, 150 K, and 300 K corresponding to paramagnetic-like behavior.

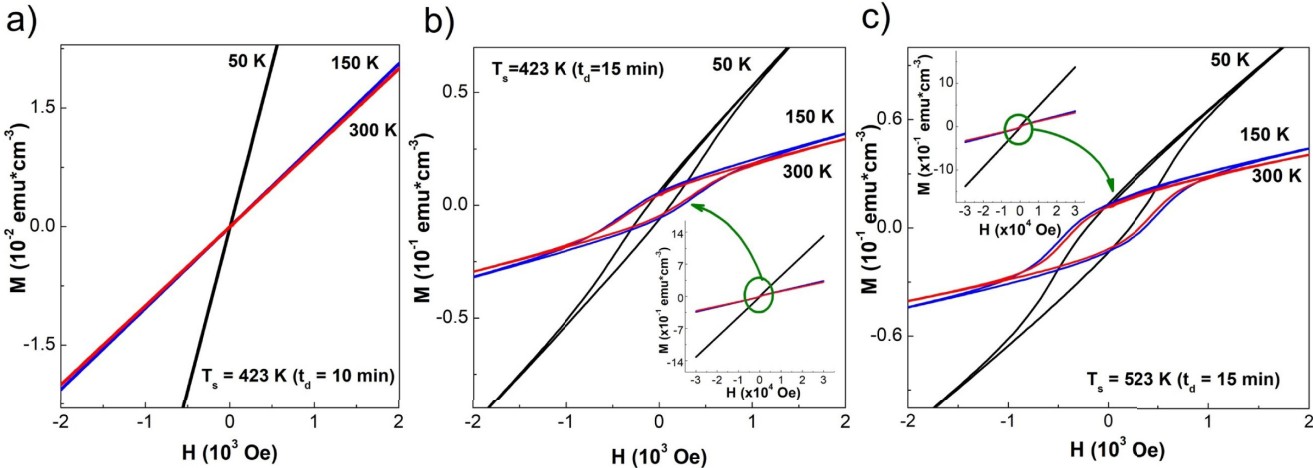

**Fig 3.** M vs. H curves at 50, 150, and 300 K for samples deposited with synthesis parameters a) T$_s$ = 423 K (t$_d$ = 10 min), b) T$_s$ = 423 K (t$_d$ = 15 min), and c) T$_s$ = 523 K (t$_d$ = 15 min). The inset of figure b) and c) highlight the magnetization response when the applied magnetic field was varied from -30000 to 30000 Oe.

In contrast, Fig 3B and 3C present hysteresis loops for the samples with t$_d$ = 15 min and substrate temperatures at 423 K and 523 K, respectively; these are associated with a ferromagnetic-like behavior. In this case, the remnant magnetization (M$_r$) and coercive field (H$_c$) values are reported in Table 1. In addition, the inset of Fig 3B and 3C shows the overall magnetization behavior when H takes values between −30 kOe and 30 kOe. The XRD measurements show the formation of secondary phases during the growth process. The presence of Mn$_2$Sb and Mn$_2$Sb$_2$ binary phases can be controlled by deposition time or substrate temperature, and crystallite size values increased (see Table 1). This study demonstrates that magnetic behavior is affected when the crystallite size increases; showing values greater magnetization and hysteresis existence (see Fig 3 and Table 1).

According to the literature, Mn$_2$Sb and Mn$_2$Sb$_2$ binary phases have a well-defined magnetic response [40, 41]. Then, the magnetic behavior of (GaMn)Sb thin films obtained is a convolution of the magnetic behaviors of Mn$_2$Sb, Mn$_2$Sb$_2$, and GaSb. The sample growth at T$_s$ = 423K and t$_d$ = 10 min shows a paramagnetic-like behavior at 50 K, 150 K and 300 K. This result and the absent of secondary phases suggest that Mn ions are distributed evenly across the sample, with no magnetic interaction and no formation of nanoclusters. The Mn ion distribution was corroborated with EDXS measurements at different energies of the incident electron beam. Authors reported similar magnetic behavior with single domains and mixed superparamagnetic states [42].

As the substrate temperature is increased, the apparition of a hysteretic response in the magnetic moment evidenced a ferromagnetic-like behavior (Fig 3B and 3C). For the sample with T$_s$ = 423 K and t$_d$ = 15 min, the secondary phases start to contribute to the total magnetization. For the sample with T$_s$ = 523 K and deposition time of 15 minutes, the magnetic character appears to be like the previous sample. A hysteretic magnetic behavior appears at room temperature, 150 K and 50 K. The absence of a magnetic saturation indicates a paramagnetic contribution as well.

Moreover, the samples were characterized by Hall-effect measurements, as presented in Fig 4. The results show that the samples deposited with t$_d$ = 10 min and 15 min at T$_s$ = 423 K (Fig 4A and 4B) exhibit linear behavior between the Hall resistivity (ρ$_{xy}$) and H. However, the sample with T$_s$ = 523 K presents a non-linear relation of ρ$_{xy}$ as a function of H. Furthermore, the ρ$_{xy}$ values for the sample with td = 10 min are 4 magnitude orders larger than the Hall

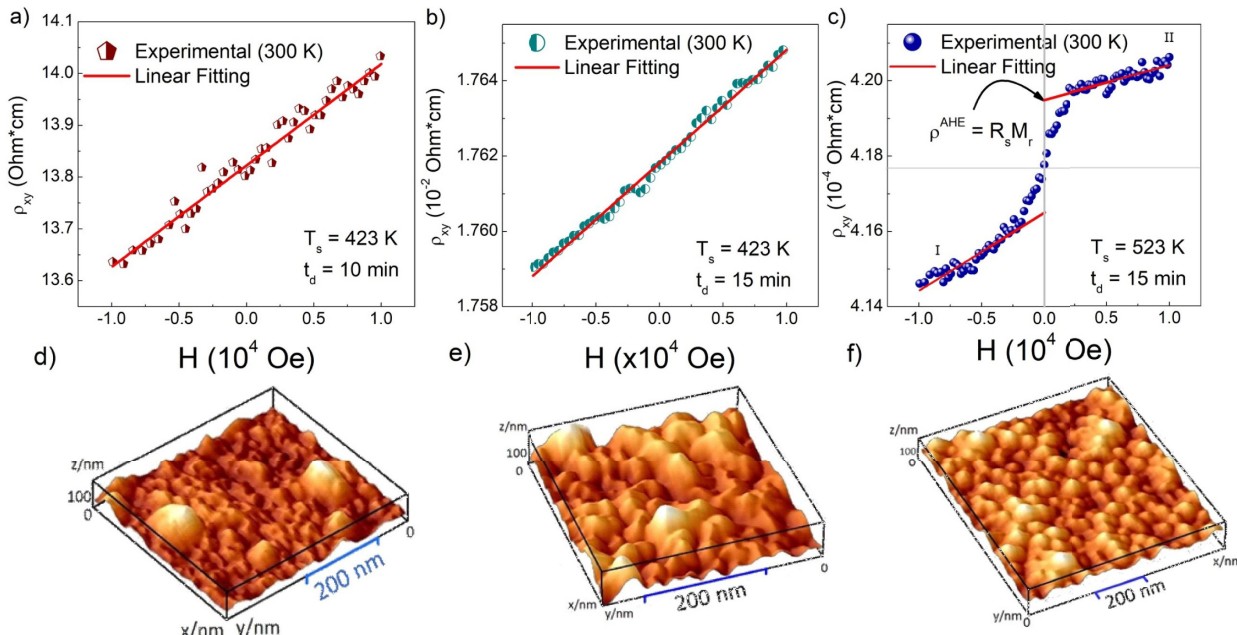

**Fig 4.** $\rho_{xy}$ as a function of H for the samples with synthesis parameters: a) $t_d$ = 10 min with $T_s$ = 423 K, b) $t_d$ = 15 min with $T_s$ = 423 K (ordinary Hall-effect), and c) $t_d$ = 15 min with $T_s$ = 523 K (Anomalous Hall-effect). AFM measurements for (GaMn)Sb thin films, with the same synthesis parameters, correspond to the pictures d), e), and f).

resistivity values for other samples, making evident that the synthesis parameters have an effect on the decrease of the Hall resistivity of the GaMnSb thin films.

Finally, the bottom panels in Fig 4 shows the surface topography obtained by AFM measurements. The synthesis parameters are correlated with the surface roughness (Fig 4D–4F: 2.21 nm, 2.56 nm, and 2.88 nm, respectively). The increase of deposition time and substrate temperature facilitate the formation of nucleation centers and their diffusion across the sample. Thus, surface morphology is characterized by different size grains.

In order to identify the magnetic contributions of each sample, additional transport and AHE measurements were performed, as shown in Fig 4. The resistivity can be expressed by the relation $\rho_{xy} = R_{Hall}^{*}H$, where $R_{Hall}$ is the Hall coefficient. Since $R_{Hall} > 0$, the majority of charge carriers, which contributed to an electrical current, are holes, this indicates that all the deposited films have a p-type semiconductor behavior. Also, the increment of the Mn in the compound increases the holes at the semiconductor material; these results have been reported by other authors [26, 43]. These results suggest that the GaSb is the main phase of the samples (Fig 1) as suggested by the XRD measurements.

In Fig 4B, a typical Hall behavior is evidenced, even though a clear ferromagnetic-like contribution appears in the M vs. H curves. The ferrimagnetic and ferromagnetic phases identified can function as independent magnetic moments and, given the granular formations on the surface (inset Fig 3), it is possible to associate the hysteresis loops with the dipolar interactions between the magnetic moments of the free Mn ions distributed in the sample, and those corresponding to the crystalline formations of Mn$_2$Sb and Mn$_2$Sb$_2$ binary phases. Additionally, the formation of these phases near the surface of the samples, corroborated from the EDXS measurements shown in Table 1, contribute to the magnetization through the surface anisotropy.

In Fig 4C, a non-linear relationship between $\rho$ and H occurs, resulting in an anomalous Hall-effect [44]. In this case, assuming that H is perpendicular to the sample surface, $\rho$ is given

by $\rho = R_{Hall}{}^*H + R_s{}^*M$, R$_s$ being the anomalous Hall coefficient, and the value of the $R_s{}^*M$ is given at Table 1. Three different R$_{Hall}$ values that generate three different values for the charge carrier concentration (n$_p$) were observed. For values of -3 kOe < H < 3 kOe, the first term of the equation presents a significant contribution to resistivity because the values of M are negligible compared to H, turning out in a carrier concentration value in the order of $10^{20}$ cm$^{-3}$ (Table 1). The carrier density of (GaMn)Sb thin films is larger than the reported for the matrix semiconductor (in the order of $10^{17}$ cm$^{-3}$) [45].

In contrast, for values of H > 3 kOe the anomalous Hall-effect is observed in the systems. It is due to that the interaction between the spins of the carriers with the magnetization of the material is the dominant contribution to the transversal conductivity. The AHE source can be associated with a nonzero total magnetization of the Mn$_2$Sb and Mn$_2$Sb$_2$ phases in the compound, and with the large spin-orbit coupling interaction in the system. In addition, the transversal anomalous Hall conductivity presents a large value of 2380 S/cm, comparable with the observed for Fe-pure and Fe-doped Mn films [46].

## Conclusions

In summary, (GaMn)Sb thin films via DC magnetron co-sputtering were obtained and the correlation between the presence of the binary phases, identified through XRD measurements, magnetic properties, and synthesis parameters was established. The compound presents a hysteresis loop at room temperature, associated with the strong magnetic dipolar interaction and surface anisotropy. In addition, M vs T curves evidence paramagnetic-like behavior to different applied magnetic fields. Variation of deposition time and substrate temperature favors the formation of Mn$_2$Sb$_2$, GaSb, and Mn$_2$Sb phases. The presence of the ferrimagnetic (Mn$_2$Sb) and ferromagnetic (Mn$_2$Sb$_2$) phases in the compound and the hysteresis processes were correlated. Through Hall-effect measurements, it was determined that most charge carriers are holes. Finally, the anomalous Hall-effect found in the sample, with higher substrate temperature and deposition time, evidence the presence of the Mn$_2$Sb$_2$ and Mn$_2$Sb phases.

## Acknowledgments

The Universidad Nacional de Colombia, COLCIENCIAS, Universidad del Rosario and Universidad del Norte supported this work. J.G.R acknowledge support from FAPA program of *Facultad de Ciencias* and *Vicerrectoria de Investigaciones* of *Universidad de los Andes*, Bogotá, Colombia. Authors acknowledge support from Microscopy Laboratory at Universidad de los Andes.

## Author Contributions

**Conceptualization:** Jorge A. Calderón, F. Mesa, A. Dussan, Juan Gabriel Ramirez.

**Data curation:** R. González-Hernandez.

**Formal analysis:** F. Mesa, A. Dussan, R. González-Hernandez, Juan Gabriel Ramirez.

**Investigation:** Jorge A. Calderón, A. Dussan.

**Methodology:** A. Dussan.

**Resources:** R. González-Hernandez.

**Supervision:** F. Mesa, A. Dussan.

**Writing – original draft:** F. Mesa.

**Writing – review & editing:** F. Mesa.

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
