## [Decision Letter · Decision Letter 0]

7 Nov 2019

PONE-D-19-27209

The effect of Mn2Sb2 and Mn2Sb secondary phases on magnetism in (GaMn)Sb thin films

PLOS ONE

Dear Dr. Mesa,

Thank you for submitting your manuscript to PLOS ONE. After careful consideration, we feel that it has merit but does not fully meet PLOS ONE’s publication criteria as it currently stands. Therefore, we invite you to submit a revised version of the manuscript that addresses the points raised during the review process.

We would appreciate receiving your revised manuscript by Dec 22 2019 11:59PM. To enhance the reproducibility of your results, we recommend that if applicable you deposit your laboratory protocols in protocols.io, where a protocol can be assigned its own identifier (DOI) such that it can be cited independently in the future. For instructions see: http://journals.plos.org/plosone/s/submission-guidelines#loc-laboratory-protocols

We look forward to receiving your revised manuscript.

Kind regards,

Sefer Bora Lisesivdin, Ph.D.

Academic Editor

PLOS ONE

Journal Requirements:

3. ** Please include your tables as part of your main manuscript and remove the individual files **. Please note that supplementary tables (should remain/ be uploaded) as separate "supporting information" files

Reviewers' comments:

Reviewer's Responses to Questions

**Comments to the Author**

1. Is the manuscript technically sound, and do the data support the conclusions?

Reviewer #1: Partly

2. Has the statistical analysis been performed appropriately and rigorously? 

Reviewer #1: N/A

3. Have the authors made all data underlying the findings in their manuscript fully available?

Reviewer #1: Yes

4. Is the manuscript presented in an intelligible fashion and written in standard English?

Reviewer #1: Yes

5. Review Comments to the Author

Reviewer #1: Review- The effect of Mn2Sb2 and Mn2Sb secondary phases on magnetism in (GaMn)Sb thin films

In this manuscript, authors describe an alternate way of achieving dilute magnetic semiconductor thin films samples. They use DC-Magnetron co-sputtering to achieve Mn doped GaSb thin films. Their magnetic measurements and anomalous Hall conductivity measurement suggests a magnetic thin film behavior. However, their argument that Mn2Sb2 and Mn2Sb secondary phases are the reason is non-existent.

Figure 1 shows their XRD measurement that authors claim to support their argument. However, there is no clear XRD response that can be observed for Mn2Sb2 and Mn2Sb. Every peak 2-theta value that can differentiate these secondary phases is missing. Authors claim to fit Mn2Sb at around 42 deg, but that peak also corresponds to GaSb. Similarly, peak at around 61 deg is identified as Mn2Sb2, but also belong to GaSb. The sample deposited at 373 K substrate temperature is clearly amorphous. And subsequent increase in substrate temperature suggest crystallization of GaSb with no signature of secondary phases.

While I agree that magnetic behavior is observed in the thin films, but mechanistic explanation is unsatisfactory. I think authors need to explore transmission electron microscope studies to verify their argument. If there are secondary phases, they should be able to see and characterize them inside TEM. Without clear proof, this work is incomplete.

Additional corrections-

There are significant grammatical errors that need to be corrected. I list a few below but authors need to do a better job of eliminating them.

Page 1- line 26: “the absent” should be “the absence”

Page 1 line 27: “or substrate temperature” should be “or higher substrate temperature”

Page 2 line 30: “as big as one order of magnitude” does not make sense. It might be “one order of magnitude larger than reported values”.

Page 2 line 32: “magnetization” should be “magnetic”. “transversal” is incorrect and not needed since anomalous Hall conductivity by definition is transverse.

Page 3 line 51: “architectures” should be “architectural”.

6. PLOS authors have the option to publish the peer review history of their article (what does this mean?). If published, this will include your full peer review and any attached files.

Reviewer #1: No

---

## [Author Response · Author response to Decision Letter 0]

24 Mar 2020

Dear Editor

Following the feedback received by reviewer, we present comments and corrections for paper titled: The effect of Mn2Sb2 and Mn2Sb secondary phases on magnetism in (GaMn)Sb thin films

All corrections were done taking the comments of the editor office.

Reviewer #1: Review- The effect of Mn2Sb2 and Mn2Sb secondary phases on magnetism in (GaMn)Sb thin films

In this manuscript, authors describe an alternate way of achieving dilute magnetic semiconductor thin films samples. They use DC-Magnetron co-sputtering to achieve Mn doped GaSb thin films. Their magnetic measurements and anomalous Hall conductivity measurement suggests a magnetic thin film behavior. However, their argument that Mn2Sb2 and Mn2Sb secondary phases are the reason is non-existent.

Figure 1 shows their XRD measurement that authors claim to support their argument. However, there is no clear XRD response that can be observed for Mn2Sb2 and Mn2Sb. Every peak 2-theta value that can differentiate these secondary phases is missing. Authors claim to fit Mn2Sb at around 42 deg, but that peak also corresponds to GaSb. Similarly, peak at around 61 deg is identified as Mn2Sb2, but also belong to GaSb. The sample deposited at 373 K substrate temperature is clearly amorphous. And subsequent increase in substrate temperature suggest crystallization of GaSb with no signature of secondary phases.

While I agree that magnetic behavior is observed in the thin films, but mechanistic explanation is unsatisfactory. I think authors need to explore transmission electron microscope studies to verify their argument. If there are secondary phases, they should be able to see and characterize them inside TEM. Without clear proof, this work is incomplete.

ANSWER:

TEM measurements were performed as requested by the reviewer. However, these measures did not give adequate information due to the nature of the samples on glass substrate. For this reason, we decided to perform complementary Raman spectroscopy measures and the results are consistent with the request. Figure 2 was included and the following text was added on line 144 through 161 (page7)

“However, Raman spectroscopy was realized in order to obtain complementary structural information to XRD measurements shown in Fig.1. Fig 2 shows the Raman spectra recorded for GaMnSb samples both, at 423 K and 523 K substrate temperature. The presence of a vibration mode for high frequency region is observed at 647 cm− 1 when Ts increases (Ts= 523 K); this can be associated to manganese oxides formation on the surface of the sample due to oxidation process after the synthesis of the material. The phonon mode observed in the Raman shift around 650 cm−1 has been reported for Mn3O4-like phase [37].

On the other hand, Raman spectra for GaMnSb samples measured show weak phonon modes at 230 cm-1, 141 cm-1 and 108 cm−1, related with LO and TO modes for GaSb and Mn2Sbx phases on the material. In the up inset of Fig 2 shows the deconvolution of the pick found at 230 cm-1 Raman shift, characterized by three weak vibrations at 226 cm-1, 220 cm-1 and 215 cm-1, which are associated to LO (226 cm-1) and TO modes, respectively. TO modes observed in the samples here, can be related with substrate inhomogeneity and imperfections on surface [38]. The presence of phonon modes observed at 141 cm-1 and 108 cm−1 can be related with the incorporation Mn atoms during synthesis process. It was reported [39] that MnSb-like vibration modes are expected in GaMnSb samples as Mn atoms have taken into GaSb matrix. The above is in accordance with the results reported in Fig 1.

Fig 2. Raman shift curves of GaMnSb thin films varying td a) 423 K and b) 523 K”

Additionally, the authors added the following reference of the discussion:

[37] Kyung-Wan Nam, Kwang-Bum Kim. Manganese Oxide Film Electrodes Prepared by Electrostatic Spray Deposition for Electrochemical Capacitors. J. Electrochem. Soc. 2006; 153: A81

[38] Zhou Xiuli, Guo Wei, Perez-Bergquist Alejandro G, Wei Qiangmin, Chen Yanbin, Sun Kai, Wang Lumin. Optical Properties of GaSb Nanofibers. Nanoscale Res. Lett. 2011; 6:6

[39] Yang Guandong, Zhu Feng, Dong Shan. Fabrication of ferromagnetic GaMnSb by thermal diffusion of evaporated Mn. J. Cryst. Growth. 2011; 316:145–148.

Additional corrections-

There are significant grammatical errors that need to be corrected. I list a few below but authors need to do a better job of eliminating them.

Page 1- line 26: “the absent” should be “the absence”

Page 1 line 27: “or substrate temperature” should be “or higher substrate temperature”

Page 2 line 30: “as big as one order of magnitude” does not make sense. It might be “one order of magnitude larger than reported values”.

Page 2 line 32: “magnetization” should be “magnetic”. “transversal” is incorrect and not needed since anomalous Hall conductivity by definition is transverse.

Page 3 line 51: “architectures” should be “architectural”.

ANSWER:

All corrections were done in the manuscript

Best Regards,

Authors.

---

## [Editor Report · Decision Letter 1]

26 Mar 2020

The effect of Mn2Sb2 and Mn2Sb secondary phases on magnetism in (GaMn)Sb thin films

PONE-D-19-27209R1

Dear Dr. Mesa,

We are pleased to inform you that your manuscript has been judged scientifically suitable for publication and will be formally accepted for publication once it complies with all outstanding technical requirements.

With kind regards,

Sefer Bora Lisesivdin, Ph.D.

Academic Editor

PLOS ONE
---

## [Editor Report · Acceptance letter]

30 Mar 2020

PONE-D-19-27209R1 

The effect of Mn2Sb2 and Mn2Sb secondary phases on magnetism in (GaMn)Sb thin films 

Dear Dr. Mesa:

I am pleased to inform you that your manuscript has been deemed suitable for publication in PLOS ONE. Congratulations! Your manuscript is now with our production department. 

With kind regards,

on behalf of

Prof. Dr. Sefer Bora Lisesivdin 

Academic Editor

PLOS ONE